# Recent Advances on P-Glycoprotein (ABCB1) Transporter Modelling with In Silico Methods

**DOI:** 10.3390/ijms232314804

**Published:** 2022-11-26

**Authors:** Liadys Mora Lagares, Marjana Novič

**Affiliations:** Theory Department, Laboratory for Cheminformatics, National Institute of Chemistry, 1000 Ljubljana, Slovenia

**Keywords:** P-glycoprotein, structure-based, ligand-based, homology modelling, molecular dynamics simulations, machine learning, computational models

## Abstract

ABC transporters play a critical role in both drug bioavailability and toxicity, and with the discovery of the P-glycoprotein (P-gp), this became even more evident, as it plays an important role in preventing intracellular accumulation of toxic compounds. Over the past 30 years, intensive studies have been conducted to find new therapeutic molecules to reverse the phenomenon of multidrug resistance (MDR) ), that research has found is often associated with overexpression of P-gp, the most extensively studied drug efflux transporter; in MDR, therapeutic drugs are prevented from reaching their targets due to active efflux from the cell. The development of P-gp inhibitors is recognized as a good way to reverse this type of MDR, which has been the subject of extensive studies over the past few decades. Despite the progress made, no effective P-gp inhibitors to reverse multidrug resistance are yet on the market, mainly because of their toxic effects. Computational studies can accelerate this process, and in silico models such as QSAR models that predict the activity of compounds associated with P-gp (or analogous transporters) are of great value in the early stages of drug development, along with molecular modelling methods, which provide a way to explain how these molecules interact with the ABC transporter. This review highlights recent advances in computational P-gp research, spanning the last five years to 2022. Particular attention is given to the use of machine-learning approaches, drug–transporter interactions, and recent discoveries of potential P-gp inhibitors that could act as modulators of multidrug resistance.

## 1. Introduction

In the last few decades, molecular modelling and computational chemistry have taken important roles in understanding the basis of drug-receptor interactions and in assisting researchers with designing new therapeutic agents [1,2]. The discovery of new drugs by trial and error is both time consuming and expensive, so computational tools are currently used in almost every step of this process, especially in the early stages [3,4]. Advances in computational chemistry algorithms and tools have driven development in this field, helping to shorten the drug design process and reduce costs. On the other hand, P-glycoprotein (P-gp) (Figure 1), an important membrane transporter expressed by the mdr1 gene and belonging to the ATP-binding cassette superfamily (ABC), has been extensively studied; nevertheless, many aspects of its structure-function mechanism remain unresolved. Recently, it has been shown [5] that P-gp is even capable of exporting relatively large peptides, such as Abeta (Aβ), not only in the BBB endothelium, but also out of neurons through an active process mediated by the membrane transporter. P-gp plays an important role in both drug absorption and drug disposition and has been implicated in the regulation of toxicity and failure of cancer therapies due to multidrug resistance (MDR), which was first discovered in the context of cancer but was later associated with other diseases, such as some autoimmune disorders and infectious diseases. P-gp confers resistance by preventing the accumulation of drugs in the cell, thus preventing their cytotoxic or apoptotic effects. This is achieved by its ability to mediate ATP-dependent translocation of drug across the plasma membrane against substantial concentration gradients. The overexpression of ABC membrane transporter proteins that actively pump drugs out of the cells is the major mechanism for MDR [6]. Modulation of MDR transporters, including P-gp, has emerged as a pharmacological target for the treatment of cancer. Agents that inhibit P-gp and related efflux proteins are expected to increase the intracellular concentrations of chemotherapeutic agents and restore their sensitivity. Therefore, it seems essential to search for computational methods that allow the development of efficient in silico screening tools that provide a rapid and cost-effective platform for the identification of potential P-gp ligands.

Identification of potential substrates and inhibitors of P-gp is of great importance for overcoming MDR, either by screening for new P-gp inhibitors that might prevent the transporter from expelling drugs out of the cell or by early identification of P-gp substrates and subsequent design of substrate properties. It is striking that although P-gp has been known for more than 30 years, and despite numerous attempts, improved and selective P-gp inhibitors have not yet been developed. This is likely due to the polyspecificity of the transporter and the lack of high-resolution structural information, which has led to limited knowledge of the molecular basis of ligand-transporter interactions.

The in silico approaches available in the field of molecular modelling can be divided into ligand-based and structure-based techniques [7]. In the first case, only information from known active compounds (ligands) is used to identify other compounds with similar properties in one or more databases [8]. Ligand-based approaches are indispensable tools when structural information about the biological target is missing or when the mechanisms of action of the molecules are not well known, including cases when the 3D structure of the target is known but the active site on the receptor is unknown. The ligand-based approach has been most commonly used, either for the structure–activity relationship (SAR) or for the quantitative structure-activity relationship (QSAR) [9,10]; but regardless of the success of ligand-based methods for drug discovery, one of their major limitations is that they do not consider the receptor’s structure during the modelling process.

In this sense, structure-based tools enable the analysis of molecular target structure and ligand–target interactions. Structure-based methods are useful for predicting the binding modes of small molecules and their relative affinities [11]. They can be used to identify ligands by high throughput docking of small molecules to a rigid target protein and scoring the hits obtained based on an implicit solvent force field. Alternatively, a low-throughput explicit-solvent technique, such as molecular dynamics (MD), can be used to characterize flexible binding sites and accurately evaluate binding pathways, kinetics, and thermodynamics. Molecular docking and MD simulations are widely used modelling techniques. However, their application in the study of P-gp has been limited by the use of homology models prior to the availability of the human structure. The high flexibility and polyspecificity of P-gp complicate the use of structure-based approaches, but their application is essential for understanding the mechanism of ligand recognition and binding.

Working with membrane proteins in general presents many challenges, ranging from lack of high-resolution 3D structures to unresolved transport cycles. Therefore, with the resolution of the first crystal structure of mouse P-gp (*m*P-gp) (PDB ID: 3G5U) [12] in 2009, the application of structure-based design to this protein became more accessible and promising for the development of this field. Moreover, in 2019, the cryo-electron microscopy (cryoEM) structure of human P-gp (*h*P-gp) was solved (PDB ID: 6QEX) [13], which raised hopes for better and more efficient development in the study of this transporter.

In this review, we provide an update on the major findings of computational studies on the membrane transporter P-glycoprotein (ABCB1) reported over the last five years.

## 2. Ligand-Based Models

P-glycoprotein (P-gp) ligand-based drug design relies on knowledge of compounds known to interact with this membrane transporter. The structure–activity relationship (SAR) [14], quantitative structure-activity relationship (QSAR) [15], three-dimensional quantitative structure-activity relationship (3D-QSAR) [16], and pharmacophore models [17] have been used to predict the activity of new compounds toward P-gp. The amount of literature addressing ligand-based approaches of P-gp is enormous and dates back several decades. Since the discovery of verapamil as an agent to reverse multidrug resistance (MDRR) [18], many SAR and QSAR studies have been published. However, in recent years, models have used larger datasets and focus more on using machine-learning approaches.

In general, the ligand-based modelling approach consists of two elements: molecular descriptors and mathematical methods for deriving predictive models, such as linear models, artificial neural networks (ANN), support vector machines (SVMs), and others. The ligand-based methods require, in the first step, the creation of an appropriate model from a training dataset. Then, the model is applied to the screening set of molecules to make predictions for the property under study [19]. 

Ligand-based methods, including fingerprint-based similarity search, 2D-QSAR, pharmacophores, 3D-QSAR, and others [20,21], are faster and relatively easy to implement compared to structure-based drug design methods; one of their main advantages is their low CPU requirements. Ligand-based approaches allow the use of generalized descriptors, features, and fingerprints, making them effective virtual screening queries; they have evolved along with advances in statistics and other machine-learning algorithms, such as regression, pattern recognition, and neural networks. However, one of their limitations is that they are based on the principles of molecular similarity, which state that structurally similar molecules should elicit similar biological responses; thus, they are limited to the chemical space used in the formulation of the models, i.e., to their domain of applicability. Efforts are still needed to develop methods that can be universally applicable [8].

In the last five years, most studies have used machine-learning approaches to build predictive models, and some also combined ligand-based machine-learning approaches with structure-based techniques such as molecular docking or MD simulations. A good example is the work by Esposito et al. [22], which used a combination of machine learning and MD simulations to predict P-glycoprotein substrates. In their study, the authors used molecular dynamics fingerprints (MDFPs) as orthogonal descriptors for training substrate classification models by machine learning. The performance of the MDFPs proved to be as good as the performance of the commonly used 2D molecular descriptors, which achieved high accuracy on chemically diverse subsets. However, when challenging the models with external validation sets, only the models trained on MDFPs or property-based descriptors could be applied to regions of chemical space not covered by the training set. The work of Esposito et al. involved one of the largest datasets found in the literature for building P-gp activity models, covering 3930 compounds. On the other hand, Kadioglu et al. [23] used machine-learning approaches such as k-nearest neighbors (kNN), neural network, random forest (RF), and support vector machine (SVM) in combination with molecular docking to develop predictive models for P-gp modulators. The molecular docking step was used as further validation for the best performing model, with twenty of the predicted compounds of each class docked to the human structure of P-gp [13]. The results revealed similar docking poses to those of doxorubicin and elacridar, which are substrate and inhibitor, respectively.

### 2.1. Improving Feature Selection

In addition to the combination of ligand-based and structure-based approaches, some efforts seem to have been performed recently-first, to optimize the methodology for selecting features to build the models, and second, to find a mechanistic interpretation that could lead to future optimization of the structures. In the work of García et al. [24], the use of a boosting feature selection method is proposed to improve the performances of P-gp classifiers and to avoid the not uncommon problem that the performance of a model built with the selected subset of features is worse than the performance obtained with all features. The authors used decision trees and support vector machines (SVM) to build P-gp inhibitor and substrate models and proved that the boosting feature selection method performed better compared to standard feature selection algorithms. The dataset used in this study consisted of 1935 P-gp inhibitors and 484 substrates. To obtain informative structural rules for the analyzed endpoint, a recent study by Wang et al. [25] developed an online decision tree-based prediction server for P-gp substrates and inhibitors called *PgpRules*, which provides two separate prediction services for P-gp substrates and inhibitors. Models were built with the classification and regression trees (CART) algorithm employing fingerprints and traditional molecular descriptors; the dataset used consisted of 925 P-gp substrates and 2056 P-gp inhibitors. The performances of the generated models were good, providing classification accuracy of over 0.70 for both endpoints, but the novelty of this server is that it annotates the rules with the key structural features for the endpoint, providing a guide for structural changes in the optimization process of drug development. Similarly, a SMILE-based classification model by Prachayasittikul et al. [26] developed using the CORrelation And Logic software (CORAL), enables the discovery of important chemical features that may contribute to the inhibitory activity of the compounds. The model, built using a dataset of 2254 compounds along with SMILES attributes instead of traditional molecular descriptors, exhibited acceptable predictive performance with accuracy, sensitivity, and specificity values greater than 70% and an MCC value greater than 0.6 for training, calibration, and validation sets.

Many of the models found in the literature mainly use fingerprints or molecular descriptors as features to build the models. Recently however, Hinge et al. [27] decided to develop a binary classification model using descriptors of a new type, namely, solvation free-energy descriptors, to prove that the molecular solvation free energy theory can be used to successfully identify P-gp inhibitors. The authors used various machine-learning approaches to build the models, such as gradient boosting machines (GBM), generalized linear models (GLM), support vector machines (SVM), and weighted κ-nearest neighbors (weighted-kNN). Among these, the SVM classification model showed the best performance, employing a combination of ten three-dimensional reference interaction site model with the Kovalenko-Hirata closure approximation (3D-RISM-KH) solvation free-energy descriptors, along with other thirteen 2D descriptors. The dataset used in this study consists of 1274 compounds derived from the work of Broccatelli et al. [28], which has been used several times to construct new models. The study demonstrates that the combination of 3D-RISMKH-based descriptors with 2D descriptors increases the accuracy of the model in predicting P-gp inhibitors compared to previous classifiers, all the way to 95.6–96.9%.

### 2.2. Reducing Heterogeneity in the Data

Other studies have focused on collecting more homogeneous datasets for the prediction of the P-gp efflux ratio, because in many of the previous studies, the collected data came from different assay conditions (e.g., different cell lines), resulting in a high degree of data heterogeneity. Following this line, Ohashi et al. [29] developed regression models to predict the value of P-glycoprotein mediated efflux and classification models to predict P-gp-mediated transport potential (low, medium, or high potential), based on a dataset of 2397 compounds collected under the same experimental conditions; data were obtained using an in vitro assay developed in-house and presented in the same study. The authors built five classification models using various machine-learning methods, such as random forest, support vector machine, artificial neural network, k-nearest neighbors, and Adaboost; and the random forest method had the best performance in both regression and classification models. This model provides information on whether the compound is a strong or weak P-gp substrate, which is an advantage over the usual binary models. Chen et al. [30] also developed a predictive model for the P-gp substrate efflux ratio but using a small dataset of 63 compounds. The authors used the hierarchical support vector regression (HSVR) method, and although the dataset came from multiple literature sources, it was carefully curated to select experimental data under the same assay conditions. The predictions by HSVR showed high accuracy and agreed well with the observed experimental values. Another recent contribution to the prediction of P-gp efflux potential was the work of Watanabe et al. [31], who developed an in silico prediction model for P-gp efflux potential in brain capillary endothelial cells (BCEC). The authors built three predictive models for the in vitro P-gp net efflux ratio using their own dataset and publicly available data. The model was constructed using the gradient boosting (GB) machine-learning method, and the proposed model was validated with new experimental brain-penetration data of 28 P-gp substrates. It showed good predictive accuracy compared with previous similar models.

### 2.3. Three-Class Classification Models

Recently, Mora Lagares et al. [32] developed a classification model that provides a qualitative prediction of P-glycoprotein inhibition/substrate activity, as it is a three-class classification model that, unlike the currently available classifiers, is able to distinguish whether the molecule under study is a substrate, inhibitor, or non-active compound. The model was developed using a counter propagation artificial neural network (CP ANN) based on a set of 2D molecular descriptors and an extensive dataset of 2512 compounds. The P-gp activity model provided good classification performance and was implemented in the online platform VEGAHUB [33], which is freely available to the public at https://www.vegahub.eu/portfolio-item/vega-qsar/ (accessed on 1 November 2022).

### 2.4. Models Including Other Transporters

Other recent studies not only aimed to predict P-gp activity but also involved the prediction of other transporters; e.g., Estrada-Tejedor et al. [34] used modified self-organizing maps (SOM) to predict drug resistance related to P-gp activity and other transporters, such as MPR1 (ABCC1) and BCRP (ABCG2). The dataset used consisted of 1204 compounds. The authors used a strategy that combines a new clustering algorithm with SOM, called consensus self-organizing maps (CSOM), to build a multi-labelled unsupervised classification model, and then compared its performance with a k-NN classifier. The performance of the model was similar to those of conventional supervised machine-learning algorithms. However, the improvement in the accuracy of substrate classification, the main advantage of CSOM, relies on its ability to identify those substrates that are more likely to be misclassified and discard those examples, thereby reducing uncertainty. On the other hand, Namasivayam et al. [35] developed a computer-aided pattern analysis (C@PA) method to discover new inhibitors for several ABC transporters, such as P-gp (ABCB1), MRP1 (ABCC1), and BCRP (ABCG2). Based on experimental data collected from 93 reports between 2004 and 2021 evaluating small inhibitors for all three transporters, the authors sought to identify the critical fingerprints for triple inhibition of ABCB1, ABCC1, and ABCG2; and the structural features that must be present for promiscuity toward the three transporters. The dataset of 1049 compounds was divided into eight classes based on their activity profiles toward the transporters. C@PA included identification of basic scaffolds, substructure search and statistical distribution, and extraction of novel scaffolds to screen a large virtual compound library. As a result of screening a public library of drug-like compounds, 45,000 hits were found for novel, broad-spectrum ABC transporter inhibitors, of which 23 were selected for biological evaluation and 5 were found to be novel lead molecules as triple ABCB1, ABCC1, and ABCG2 inhibitors.

## 3. Structure-Based Approaches

In recent years, significant improvements in proteomics and chemical genomics [36], protein chemistry, structure elucidation, and refinement techniques have led to an exponential increase in the number of available three-dimensional (3D) structures of proteins. Structure-based drug design approaches depend on knowledge of the 3D structure of the biological target; therefore, the greater availability of 3D structures of proteins has led to an increase in the application of this method. The 3D structure of the target can be obtained by methods such as X-ray crystallography, NMR spectroscopy, or cryo-electron microscopy (cryoEM) [37]. However, if no experimental structure of the target is available, it is also possible to build a homology model of the target based on the experimental structures of related proteins. 

The main advantage of structure-based methods is the ability to explore new structural prototypes and large virtual libraries in a relatively short time. They serve as good filters to remove inactive compounds or to distinguish between more and less active compounds. Nevertheless, there are some limitations associated with these techniques; for example, not all hit compounds can be present in a physical compound collection, which slows down the possibility of evaluation in biochemical assays; some molecules cannot be readily synthesized. When structural information is available, de novo generation and molecular docking can be used either as alternatives to, or in parallel with, conventional high-throughput screening methods to predict which compounds will have affinity for a given target [19].

In the study of P-gp, various structure-based approaches have been used to classify and understand ligand–P-gp interactions. Prior to the publication of the *m*P-gp structure in 2009 [12], structural studies of *h*P-gp relied heavily on homology models based on bacterial transporters [38]. Access to an X-ray structure of P-gp, albeit not at perfect resolution, was a major advance for structure-based studies of this transporter. In recent years, the number of available 3D structures of ABC proteins [39] and the power of experimental approaches have facilitated the application of structure-based methods to predict ligand–transporter interactions.

### 3.1. Homology Modelling and Molecular Docking Studies Involving In Vitro Assays

Although the structure of human P-gp [13] has been available since 2019, there are still many recent studies that relied on the use of homology models, often in combination with molecular docking and MD simulations. Some of the studies created homology models of *h*P-gp using multiple templates to improve their quality. For example, the work of Mora Lagares et al. [40] presented the construction of several homology models of *h*P-gp based on different available crystal structures of *m*P-gp and C. elegans P-gp (PDB ID: 4F4C) [41], their evaluation and subsequent use for molecular-docking calculations for a set of thirteen compounds. The results of the in silico approach to the study of ligand–P-gp interactions were consistent with the available experimental P-gp efflux results reported in the same study and with the available crystallographic literature, as many of the experimentally observed interacting key residues were also found in the in silico study. Similarly, Marques et al. [42] proposed a homology model of *h*P-gp based on several crystal structures of transporter proteins and refined the model by MD simulations. From the MD results, a conformational ensemble was selected to dock lignan compounds previously reported as P-gp inhibitors, and the results were used to evaluate binding modes, binding energies or dissociation constants as a measure of binding affinity. The authors then screened a library of 76 flavonoids, and the 10 highest-scoring compounds were selected for in vitro testing. Two of the predicted active flavonoids increased doxorubicin accumulation and cytotoxicity in resistant cancer cells (HL60/MDR), demonstrating the potential of molecular modelling and virtual screening for prioritizing candidates in drug development.

In the study by Żesławska et al. [43], it was shown that the derivatives of rhodamine have a good inhibitory effect on the P-gp efflux pump. One of the most promising compounds tested had a synergistic effect in combination with doxorubicin. Molecular docking studies showed that a carboxylic group can successfully interact with residues of the P-gp binding site despite ionization at physiological pH, thereby contributing to the binding of the inhibitor.

Some studies have searched for new MDRR agents in plant-derived compounds using in silico tools to speed up the process and better select candidates that can then be synthesized and tested in vitro. This is the case with the work of Sagnou et al. [44], who performed MD simulations and docking of some curcumin derivatives into a P-gp model based on crystal structures of *m*P-gp, and selected the derivatives with better performance/binding affinity for synthesis and evaluation of sensitization of multidrug-resistant (MDR) cells (K562 cell line) to doxorubicin. Two of the curcumin derivatives significantly increased the sensitivity of the resistant cells to doxorubicin. In addition, the compounds exhibited good water solubility, suggesting that they represent promising scaffolds for the development of clinically useful P-gp inhibitors. In the same way, Syed et al. [45] created a homology model of *h*P-gp using the 3D structure of C. elegans P-gp (PDB ID: 4F4C) [41] as a template, and performed molecular docking and MD simulations of piperine analogs. Since the in silico results were promising, the analogs were synthesized and tested in two different cancer cell lines overexpressing P-gp (KB (cervical) and SW480 (colon)). The results showed that the analogs increased the sensitivity of resistant cells to chemotherapeutic agents such as paclitaxel and vincristine compared to piperine. Furthermore, Bortolozzi et al. [46] synthesized and evaluated new ecdysteroid derivatives as P-gp inhibitors in two different resistant cell lines (CEM^Vbl100^ and LoVo^Doxo^) and one cancer cell line (medulloblastoma). The authors used the molecular docking approach to elucidate the interactions of the active ecdysteroid derivatives with the receptor by using a homology model of *h*P-gp based on the crystal structure of *m*P-gp (PDB ID: 3G60) [12]. Two derivatives showed the ability to resensitize the resistant cell lines and reduce the cancer stem cells in the medulloblastoma cell line. The in silico part of the work provided a rationale for the biological results and showed that the preferred binding site for the most active compounds was consistent with the literature on steroid binding to P-gp.

Recent studies are mostly aimed at developing new compounds that can inhibit the membrane transporter and therefore be used as novel MDRR agents, or at evaluating the relationships between new promising drug compounds and P-gp, as in the work by Pang et al. [47], where in addition to in vitro assays on Caco-2, MDCK, and MDCK-MDR1 cells, molecular docking simulations were performed to investigate the corresponding P-gp–ligand interactions of DL0410, a novel drug for the treatment of Alzheimer’s disease, using a homology model of *h*P-gp based on the crystal structure of *m*P-gp (PDB ID: 4KSB) [48]. The in silico study confirmed the in vitro results and suggested that DL0410 is a substrate and a competitive inhibitor of P-gp, as it can bind simultaneously in the presence of rhodamine but not in the presence of verapamil. In addition, some authors have attempted to improve the available in vitro assays to evaluate P-gp transport using the support of computational approaches. Hosseini Balef et al. [49] created a homology model based on the crystal structure of *m*P-gp (PDB ID: 3G61) [12], stabilized it by MD simulations and used it for flexible docking of 80 drugs. This study demonstrated the relationship between MD simulations and flexible docking with cellular experiments using a ^99m^Tc-MIBI radiotracer to evaluate potential P-gp inhibitors. The results show that the cell-based radiotracer assay is accurate and rapid for screening compounds in the search for P-gp inhibitors, but the authors also concluded that lipophilicity and molecule size are fundamental factors affecting the location of the binding site.

### 3.2. Molecular Dynamics Simulations

MD simulations have benefited from advances in recent years and have proven to be a valuable resource for the study of membrane proteins such as P-gp. In general, this approach replaces the static model with a dynamic model in which the system is set in motion. The motion is simulated by solving Newton’s dynamic equations. First, a computational model of the molecular system is created from NMR, crystallographic, cryoEM, or homology modelling data. Then, the forces acting on each atom by all other atoms in the system are estimated and inserted into Newton’s equations of motion to predict the spatial position of each atom as a function of time. These equations are solved iteratively for each particle in the system to calculate the forces acting on each atom, and then, these forces are used to update the position and velocity of each atom. The result is a 3D trajectory that describes the atomic level configuration of the system at each point during the simulation time [50]. As MD simulations can predict how each atom in a protein or other molecular system will move over time, they can capture important biomolecular processes with a temporal resolution that is not possible experimentally [51], including conformational changes, ligand binding, and protein folding. These simulations can also predict how biomolecules respond at the atomic level to perturbations such as mutation, phosphorylation, protonation, or the addition or removal of a ligand.

The application of MD in drug discovery is emerging as an important tool to understand the physical basis of the structure and function of biological macromolecules. The previous view of proteins as relatively rigid structures has been replaced by a dynamic model in which internal motions and the resulting conformational changes play an essential role in their function. Therefore, MD simulations play a key role in understanding the mode of interactions of potential novel P-gp inhibitors (MDRR agent) with the membrane transporter.

The work of Shahraki et al. [52] is a good example of what has just been said, as the authors used MD simulations to further investigate the possible binding sites of the most active 5-oxo-hexahydroquinoline derivatives according to the in vitro P-gp-mediated multidrug resistance reversal assay. The results from MD show that the active compound could be stabilized by the formation of H-bonds with some protein residues and by hydrophobic and arene-π interactions. The simulations were performed using a homology model of *h*P-gp based on the structure of *m*P-gp (PDB ID: 3G5U) [12]. Similarly, Mukhametov et al. [53] performed MD simulations to generate various conformational structures of P-gp, which were then used for docking 31 compounds. Interestingly, all experimentally confirmed substrates in the battery of compounds studied bind to the main binding cavity at the top of the protein, whereas non-substrates bind to different binding sites located mainly in the middle part of the protein structure. Compounds that experimentally showed intermediate activity toward P-gp (efflux ratio greater than 1 and less than 2) were almost equally likely to bind to both binding sites.

Since MD simulations have been widely used to study the P-gp transport mechanism and the force fields used in the simulations differ in terms of parametrization and validation because they are operated with different data and observables, Wang et al. [54] investigated the influence of the force field in MD simulations of P-gp and compared the conformational ensembles obtained after 500 ns simulations with five different force fields: AMBER 99SB-ILDN, CHARMM 36, OPLS-AA/L, GROMOS 54A7, and MARTINI. The results showed differences in the conformation of the ensembles with very little overlap, indicating that the limitation of the force field used should be taken into account in the simulation and interpretation of the results obtained.

#### 3.2.1. Role of Membrane Lipids in the Efflux Process

In addition to the study of ligand-protein interactions, MD simulations have also been used to study the conformational changes associated with the translocation process during the efflux of substrates, and more recently, interest has also been given to the study of the interactions of the lipid molecules in the lipid bilayer and their influence on the overall transport mechanism. For instance, Barreto-Ojeda et al. [55] performed coarse-grained MD simulations to investigate possible lipid access pathways in P-gp based on the crystal structure of *m*P-gp (PDB ID: 4M1M) [56]. The authors found that lipid occupancy in the P-gp cavity was low, at only about 30% of the simulation time, and that up to four lipid molecules can occupy the cavity simultaneously for very short periods (less than 0.05% of the simulation time). The simulation led to the conclusion that only lipids of the lower leaflet are involved in lipid-binding events and allowed the identification of important lipid-binding residues at the portals and in the P-gp cavity. Similarly, using a homology model of *h*P-gp inserted into a lipid bilayer that mimics the composition of brain epithelial cells, Domicevica et al. [57] performed multi-scale MD-simulations to analyze the diffusion and distribution of lipids in the system, which behaves similarly to other mixed-lipid systems, including cholesterol movement. In addition, Thangapandian et al. [58] performed the first study in which translocation of cholesterol was observed at the surface of P-gp, representing a new passive mechanism by which P-gp might contribute to the redistribution of lipids in the membrane. The authors performed a series of 40 independent MD simulations of P-gp (PDB: 4M1M) embedded in cholesterol-rich lipid bilayers to obtain a comprehensive sample of cholesterol–protein interactions in P-gp. The results revealed distinct cholesterol binding regions in P-gp, a preference for binding via the rough β-face and one complete and two partial cholesterol flipping events between the two leaflets of the bilayer mediated by the surface of P-gp.

Other studies have focused on studying the interactions of known P-gp-interacting compounds in the binding pocket while analyzing the involvement of membrane lipids in the inhibition or efflux process. Some examples of this include the work of Kapoor et al. [59], who performed MD simulations of P-gp (PDB: 4M1M) in the presence of a high-affinity inhibitor, tariquidar, and compared the results with the nucleotide-free and ATP-bound state of the protein. The authors proposed a novel inhibitory mechanism for tariquidar in which lipids appear to enhance its inhibitory effect. The inhibitor facilitates the recruitment of lipid molecules into the P-gp lumen through the two drug entry portals: the lipid heads enter the portal and the tails remain in the lipid environment. This conformation of the lipids likely reinforces the conformational restriction of the TMD already induced by the inhibitor. The work of Behmard et al. [60] employed conventional MD simulations, binding free energy calculations, steered molecular dynamics, and umbrella sampling to study the interactions of two antiepileptic drugs with P-gp and their possible efflux pathway from the binding site. The authors hypothesized that the contribution of nonpolar interactions with the P-gp channel lining and with membrane lipid molecules is the main driving force impeding efflux of the two antiepileptic drugs through *h*P-gp. Simulations were performed on a homology model of the *h*P-gp in outward facing conformation based on the crystal structure of the bacterial transporter Sav1866 (PDB ID: 2ONJ) [61].

#### 3.2.2. Exploring Ligand Binding Interactions and the Binding Pocket

In a recent study by Wang et al. [62], the transport of the P-gp inhibitor verapamil was investigated using targeted MD simulations on a homology model of *h*P-gp based on the structure of *m*P-gp (PDB ID: 4M1M), and the results were compared with those of a previous work on the transport of doxorubicin [63]. The transport of verapamil and doxorubicin appears to be driven by electrostatic repulsion in the initial phase and hydrophobic interactions in the later phase. However, the residues involved in these interactions are different for the two compounds, and in particular, the residues contributing to the hydrophobic interactions have different energy compositions, which may explain the weak competitiveness of verapamil with doxorubicin in efflux transport.

Zhang et al. [64] reported the ability of P-gp to bind more than one molecule simultaneously, with three molecules being the upper limit. The authors used the crystal structure of *m*P-gp (PDB ID: 4M1M) to perform MD simulations and found that the number of substrates that can bind simultaneously depends on the size of the molecules and that drug–drug interactions within the binding pocket are beneficial for simultaneous binding. These results demonstrate the flexibility of the binding pocket and confirm the induced fit theory.

The effects of mutations at transmembrane domains (TMDs) have also been studied by MD simulations. In the work of Bonito et al. [65], an *h*P-gp homology model based on the crystallographic structure of *m*P-gp (PDB ID: 4Q9H) was used to perform the simulations with mutations experimentally associated with changes in substrate specificity and drug-stimulated ATPase activity. The authors concluded that the mutations induce repacking of transmembrane helices and alter drug-binding pocket volume and drug-binding sites, but also affect TMD-NBD (nucleotide-binding domain) contacts and interfere with signal transmission from TMDs to NBDs.

#### 3.2.3. Experimental Structure of *h*P-gp

Since the structure of *h*P-gp [13] is known, some studies have been performed using the experimentally available structure. For example, Ibrahim et al. [66] performed computational simulations using the cryoEM structure of *h*P-gp. The authors screened a database of more than 35,000 molecules, but first evaluated the in silico protocol [44] by docking four co-crystallized ligands and comparing the docking poses with the native co-crystallized pose. They then performed MD simulations for the top 10% of the screened compounds by molecular docking to calculate the corresponding binding affinities. Five of the compounds studied showed more favorable energy values compared to Taxol, indicating them as potential drug candidates.

Mora Lagares et al. [67] also performed MD simulations using the cryoEM structure of *h*P-gp, but in this case to investigate the effects of binding of different compounds on the conformational dynamics of P-gp. The results showed significant differences in the behavior of the transmembrane protein when an active or inactive molecule was in the binding pocket, such as different movement patterns that could be correlated with the conformational changes, leading to activation of the translocation mechanism.

## 4. Conclusions

The study of P-glycoprotein (P-gp) is essential for many aspects of drug development. Since its discovery and its association with multidrug resistance (MDR) more than thirty years ago, this biologically important ATP binding cassette (ABC) membrane transporter has been a target of extensive research. However, despite the progress made in recent years, especially in the field of structural biology, there are still many unanswered questions. Recent in silico studies contribute significantly to our understanding and give us an indication of how this transporter works and the nature of the conformational cycle through which compounds are pumped out of the cell, but the exact mechanisms of ligand recognition and transport are still not clear. It is likely that a multidisciplinary approach offers the best chance of answering these and related questions.

In this review, a number of alternative approaches to modelling the membrane transporter P-gp have been presented, including integrated approaches, because the complex problem of studying P-gp cannot be solved by either ligand-based or structure-based methods alone. Each approach has its own advantages and limitations that make it suitable for specific applications. Hybrid systems, however, provide valuable modelling tools that integrate the two techniques in a sequential or parallel manner. The recent ligand-based models outlined rely on large datasets and machine-learning techniques. They allow rapid screening of large molecular databases to predict their substrate or inhibition properties, but their interpretation remains at the level of substructures or general physicochemical properties, so they have recently been applied in combination with structure-based methods to provide a mechanistic interpretation. On the structure-based side, the availability of mouse P-gp crystal structures has allowed the construction of high-quality homology models for P-gp, and the recent availability of the cryoEM structure of the ABCB1 transporter has allowed the investigation of different binding modes, ligand recognition, transport mechanisms at the molecular level, and the evaluation of potential new P-gp inhibitors (Table 1). It is hoped that these strategies will evolve and potentially yield more selective and less toxic strategies that will help reduce the MDR associated with P-gp.

The increasing amount of experimental data and computational power, and the development of computational methods to predict conformations and interactions, should improve research in this area and make it easier for researchers to make progress with ABC membrane transporters such as P-gp.

## Figures and Tables

**Figure 1 ijms-23-14804-f001:**
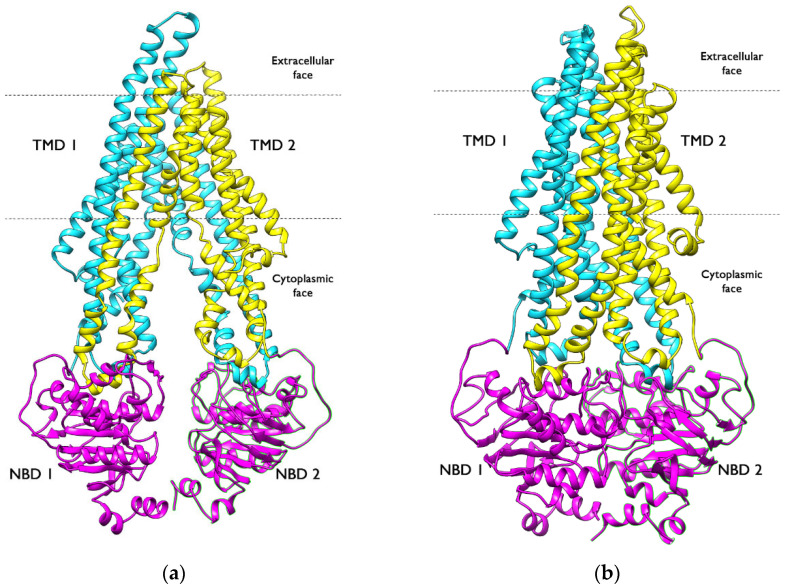
Cryo EM structures of *h*P gp. (**a**) Inward-facing conformation (PDB ID: 6QEX); (**b**) outward-facing conformation (PDB ID: 6C0V). Nucleotide binding domains (NBDs) are shown in magenta, and transmembrane domains (TMDs) are shown in cyan and yellow.

**Table 1 ijms-23-14804-t001:** Summary of potential P-gp inhibitors detected by computational studies reviewed in this manuscript.

Name	Structure	Molecular Weight	Experimental Validation	Ref.
Quinoline and 1,2,4-oxadiazole derivative 15	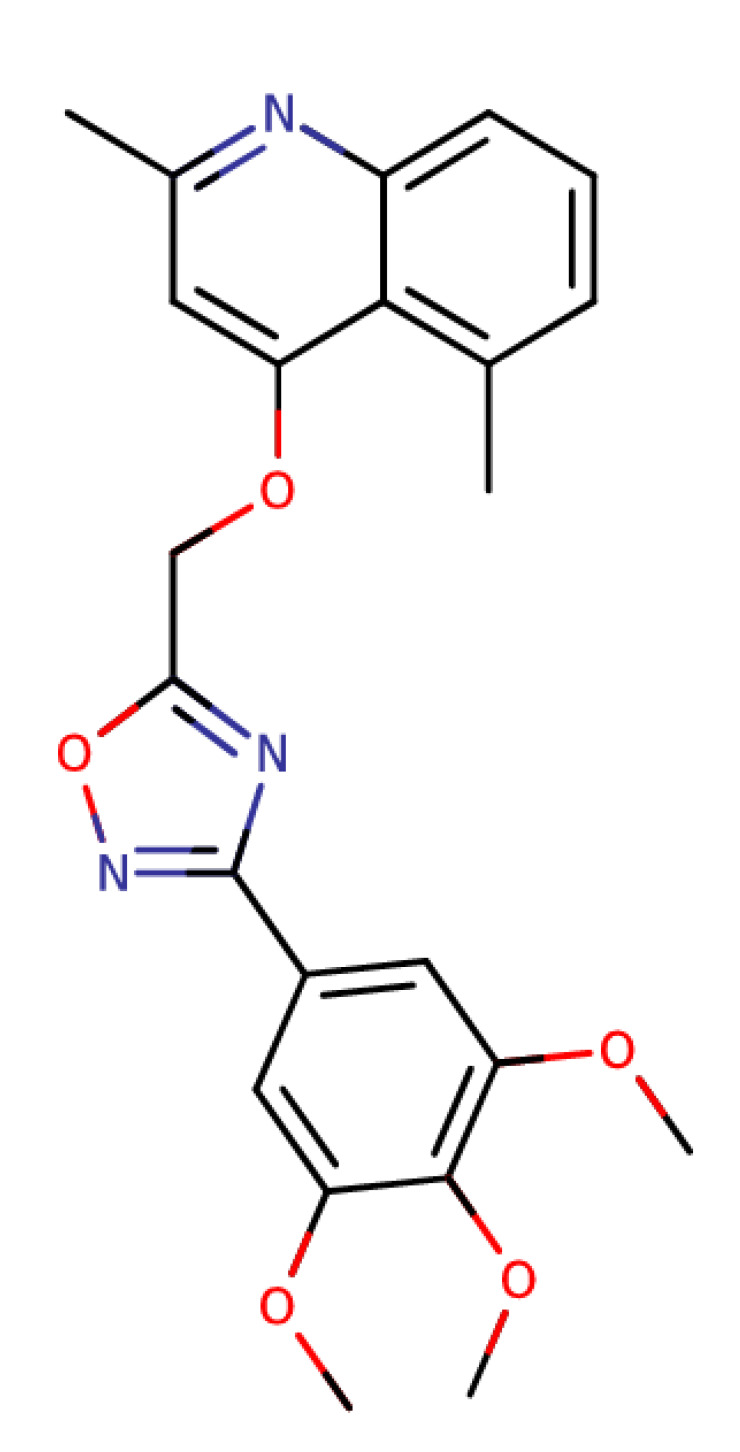	421.45	IC_50_: 8.59 μM	[35]
Quinoline 1,3,4-thiadiazole derivative 18	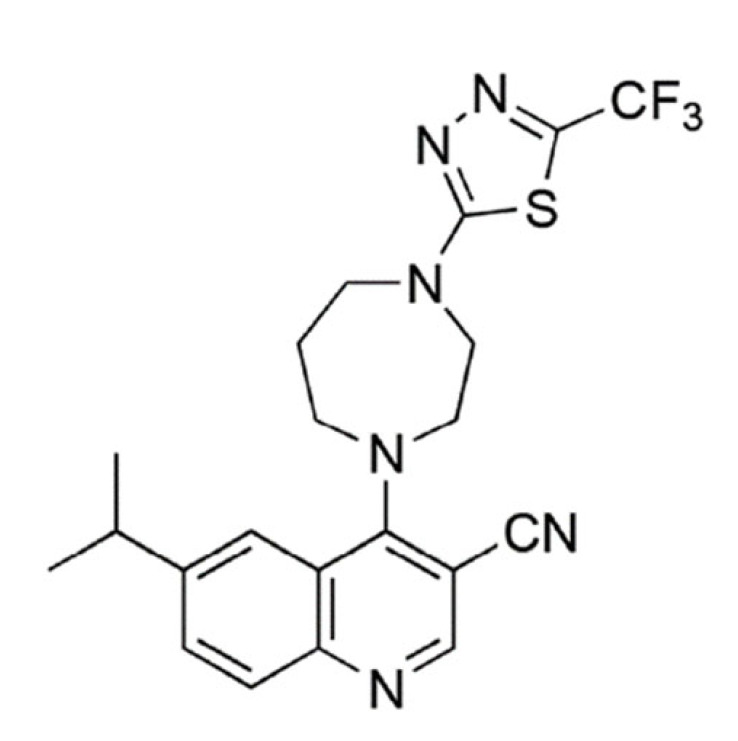	446.49	IC_50_: 2.53 μM	[35]
Quinazoline and 1,2,4-oxadiazole derivative 21	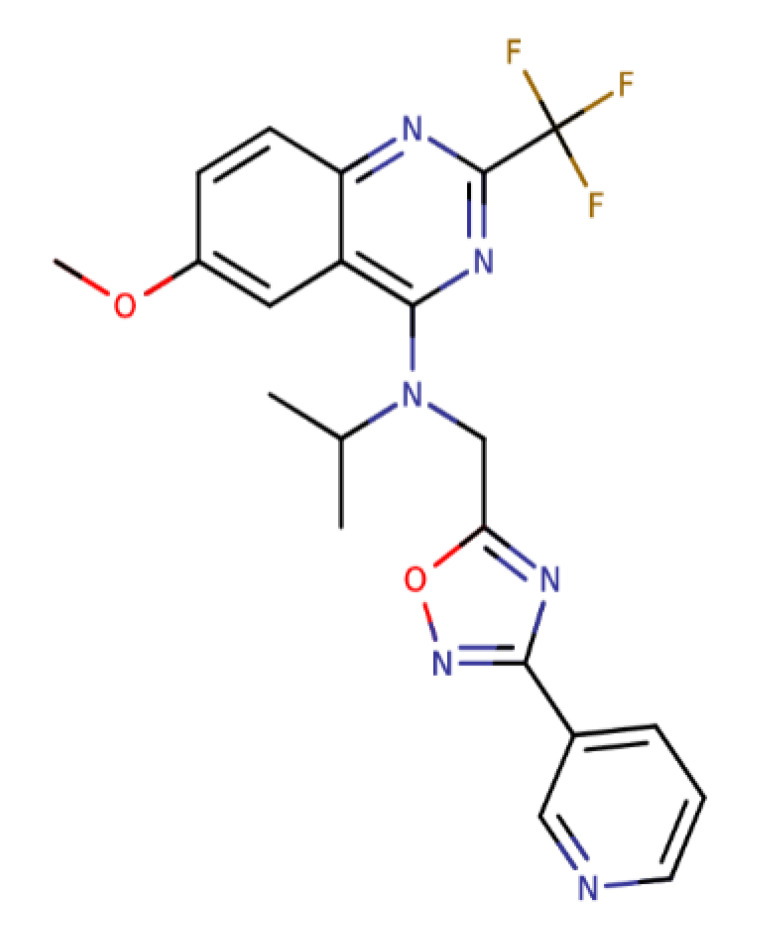	444.41	IC_50_: 2.64 μM	[35]
Quinoline and thieno [3,2-c]pyridine derivative 22	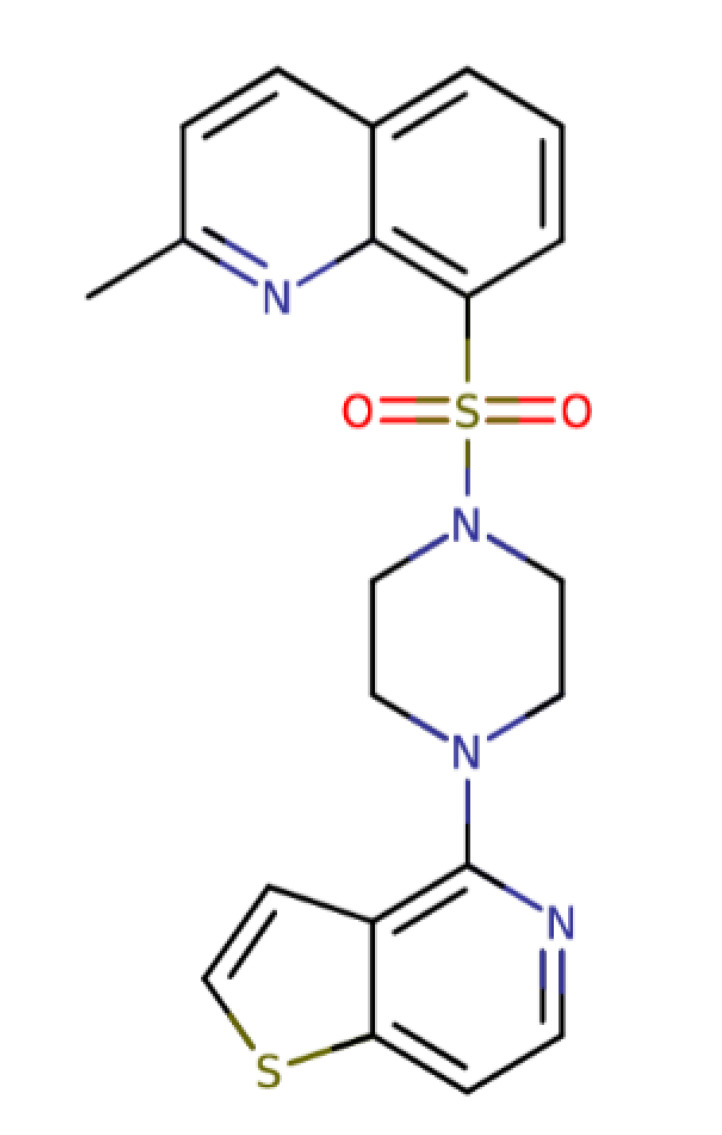	424.54	IC_50_: 3.64 μM	[35]
Quinoline and 1,2,4-oxadiazole derivative 26	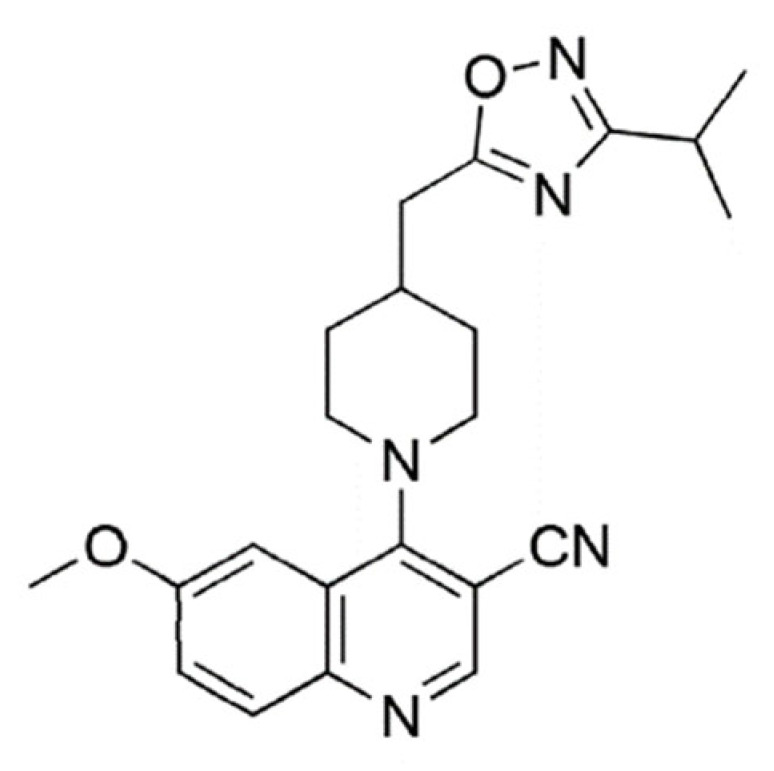	391.47	IC_50_: 2.00 μM	[35]
Baicalein	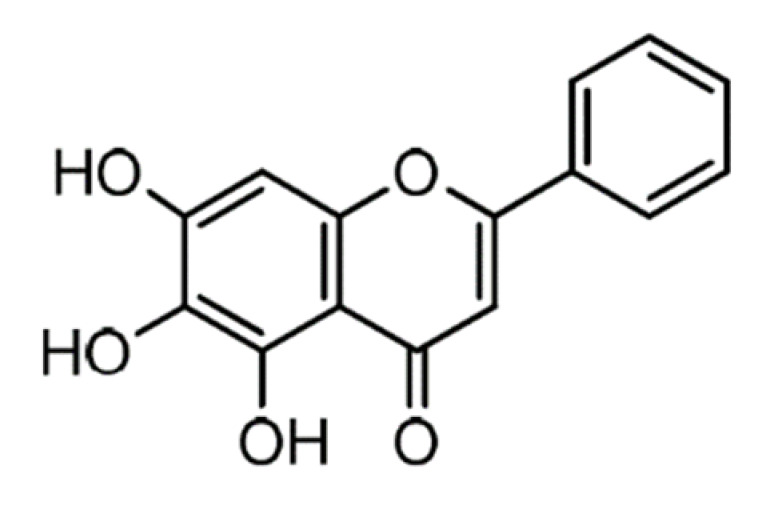	270.24	n.p. ^1^	[42]
Quercetin-3-glucoside	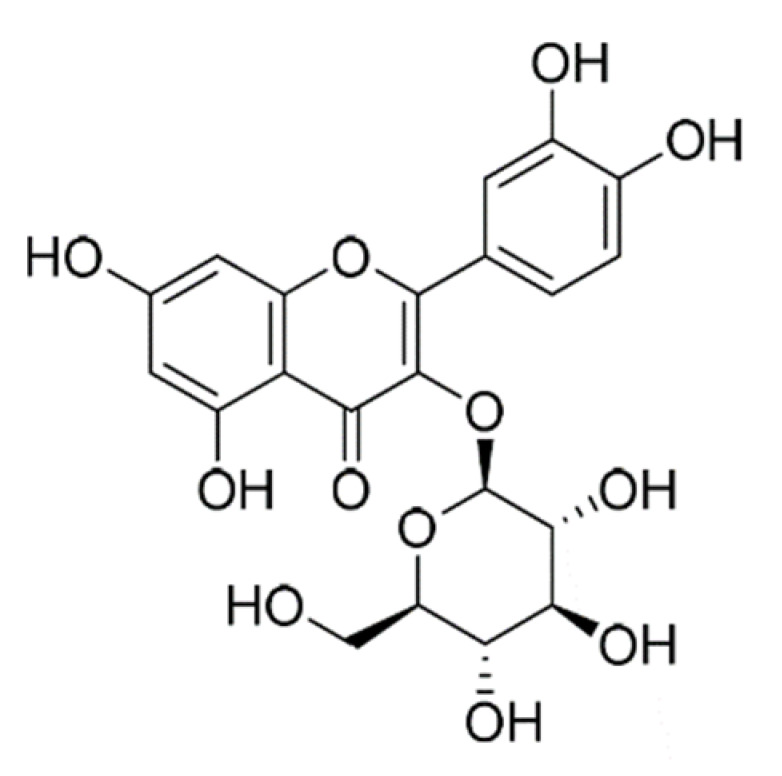	464.4	n.p.	[42]
3-(5-{[4-(diphenylamino)phenyl]methylidene}-4-oxo-2-sulfanylidene-1,3-thiazolidin-3-yl)propanoic acid	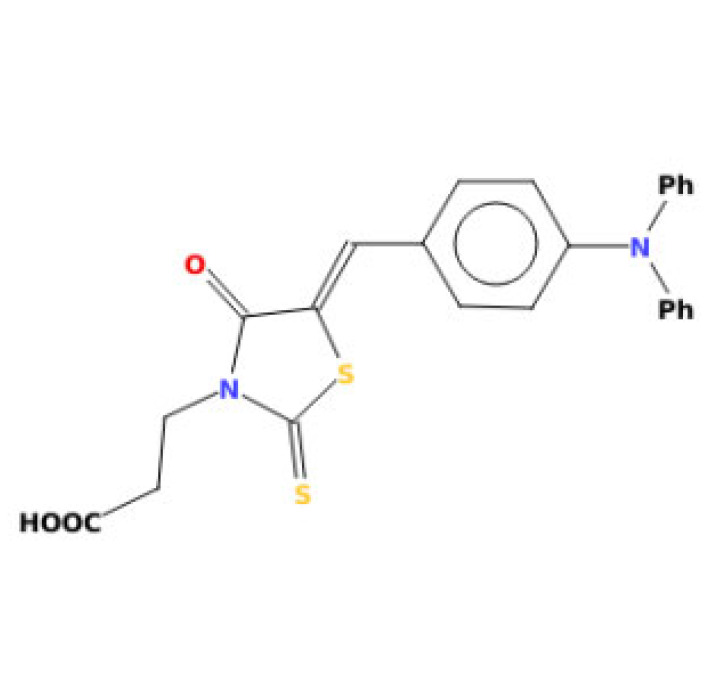	424.54	IC_50_: 42.10 μM	[43]
4-(3,5-bis((E)-3,4-dimethoxystyryl)-1H-pyrazol-1-yl)-Nethylbenzamide	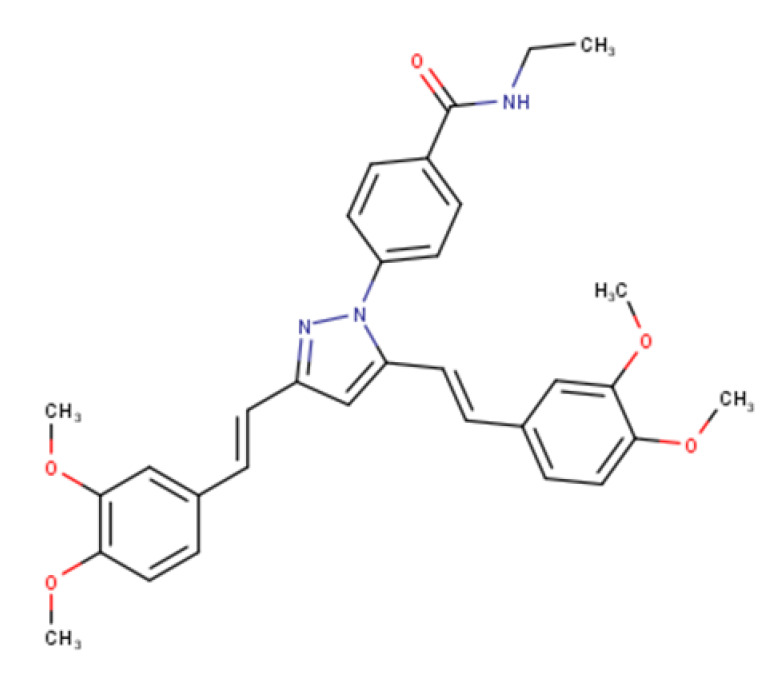	540.24	IC_50_ > 50 μM	[44]
(4-(3,5-bis((E)-3,4-dimethoxystyryl)-1H-pyrazol-1-yl)phenyl)( pyrrolidin-1-yl) methanone	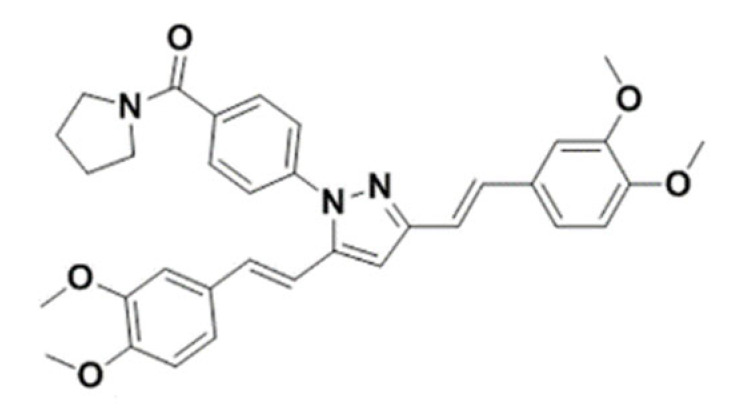	566.67	IC_50_ >50 μM	[44]
(2E,4E)-5-(benzo[d][1,3]dioxol-5-yl)-1-(6,7-dimethoxy-3,4-dihydroisoquinolin-2(1 H)-yl)penta-2,4-dien-1-one	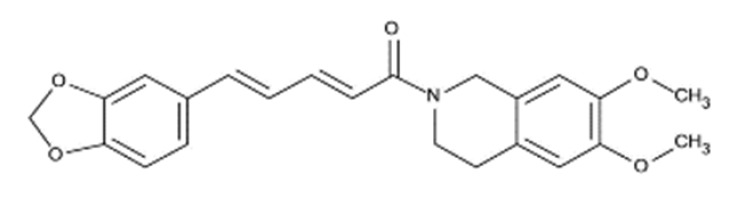	393.43	IC_50_: 2.93 nM	[45]
20-Hydroxyecdysone-2,3,20,22-dicyclohexyl-ketal	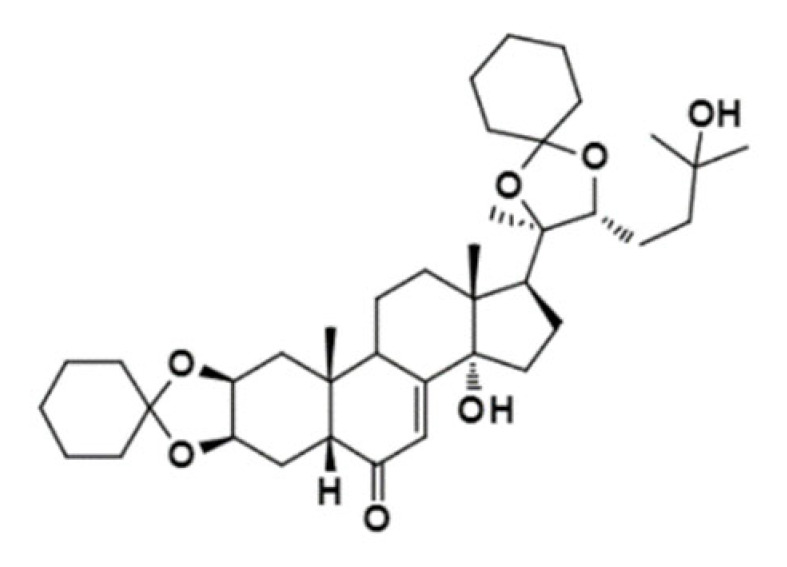	640.88	Decrease P-gp expression in the multi-drug-resistant CEM^Vbl100^ cell	[46]
20-Hydroxyecdysone 2,3,22-tribenzoate	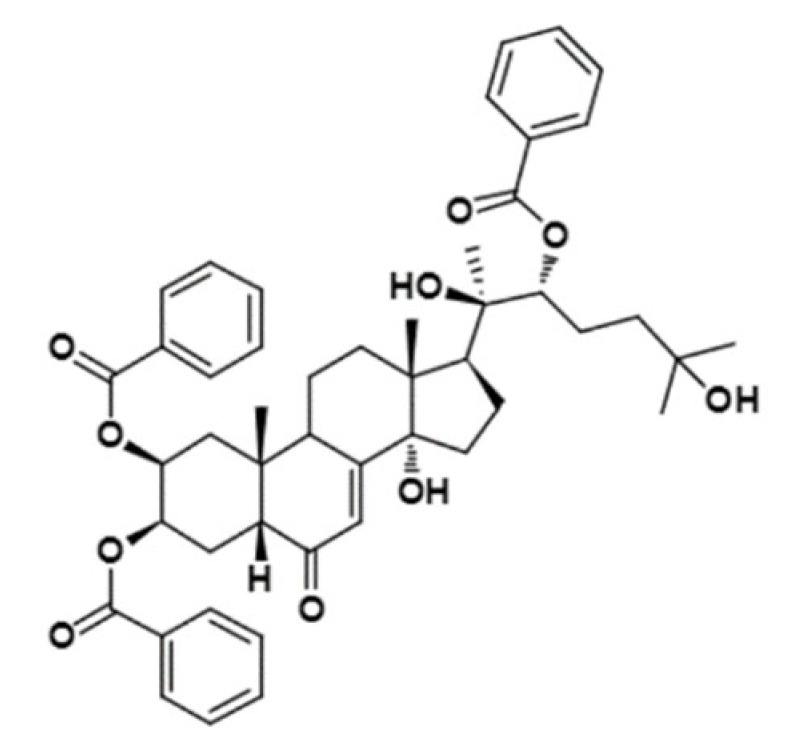	792.95	Decrease P-gp expression in the multi-drug-resistant CEM^Vbl100^ cell	[46]
DL0410	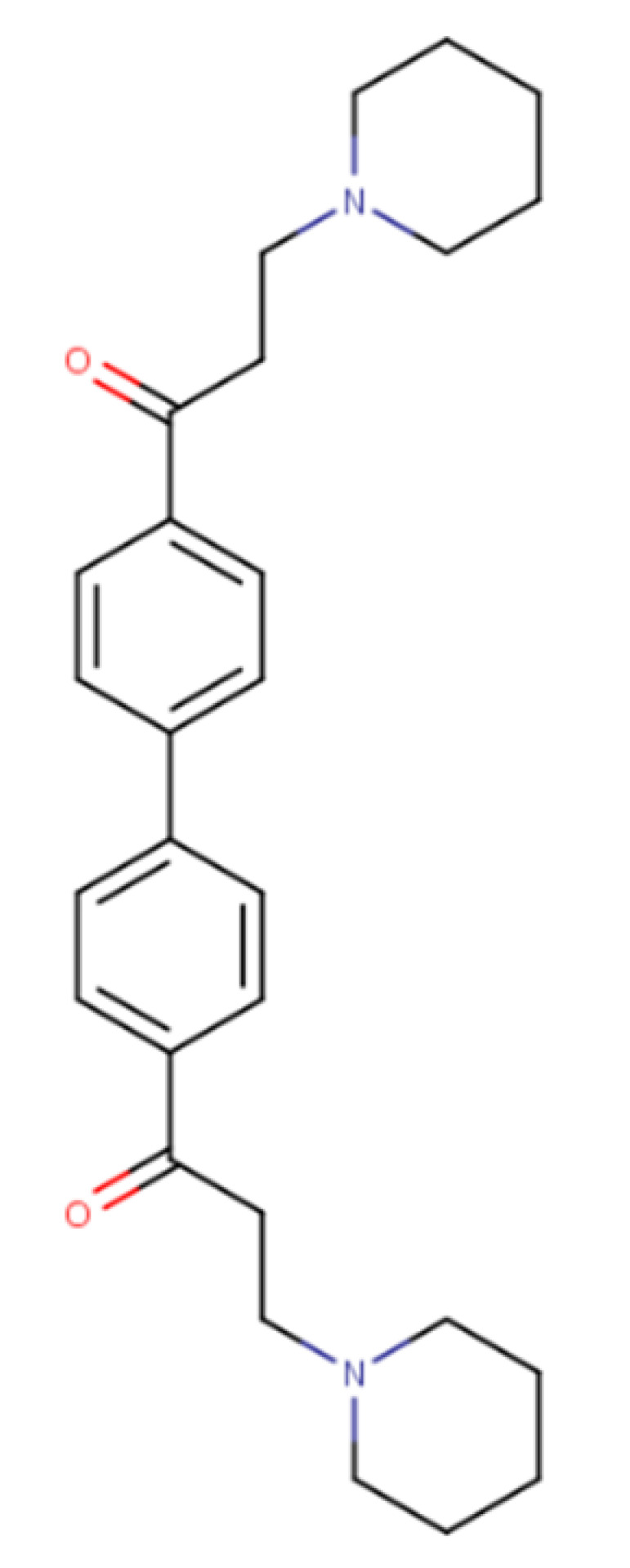	432.28	ER ^2^ (μM): 4.51	[47]
N-(4-methoxyphenyl)-2-methyl-4-(2-nitrophenyl)-5-oxo-1,4,5,6,7,8-hexahydroquinoline-3- carboxamide	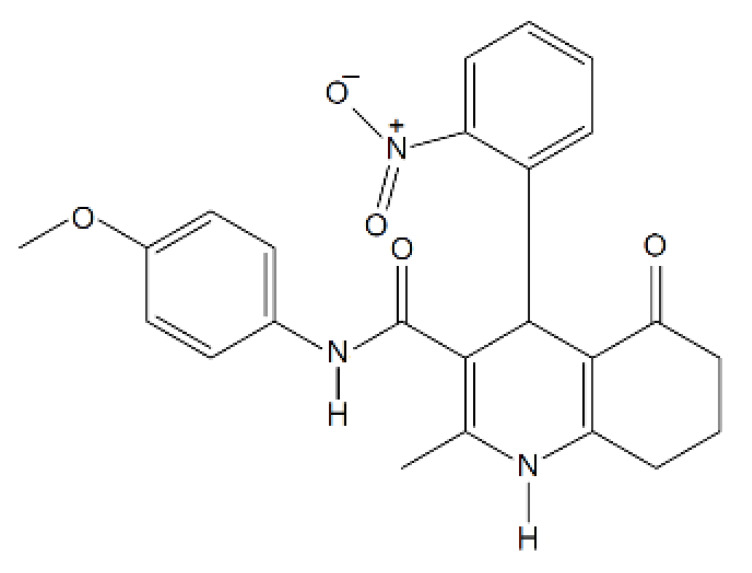	433.20	n.p.	[52]
NPC104372	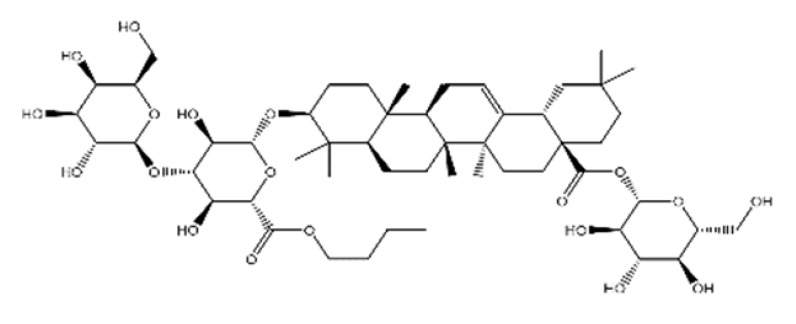	1012.56	n.p.	[66]
NPC475164	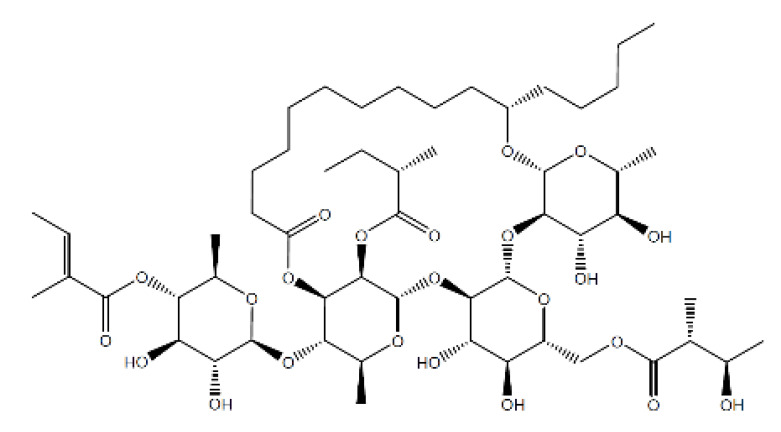	1120.60	n.p.	[66]
NPC2313	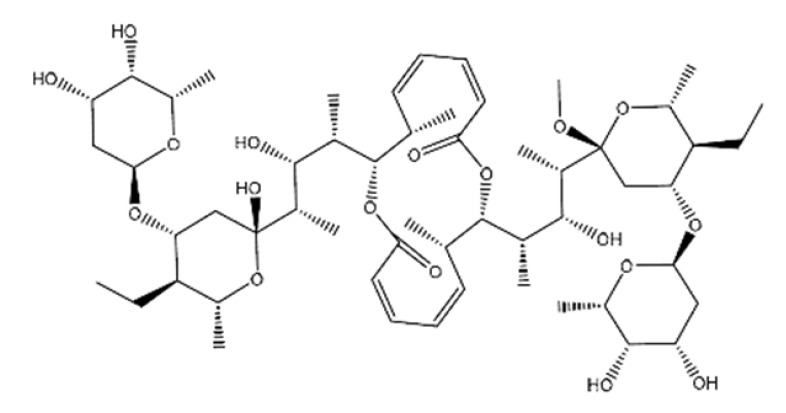	1038.61	n.p.	[66]
NPC197736	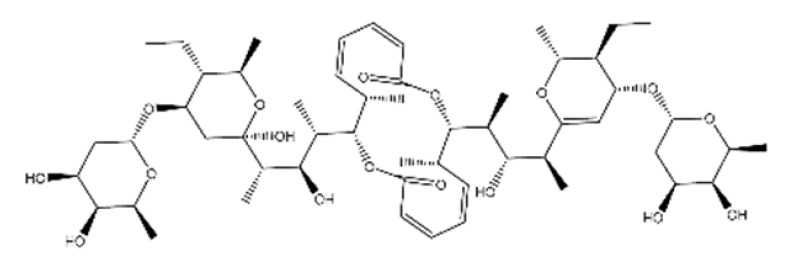	1006.59	n.p.	[66]
NPC477344	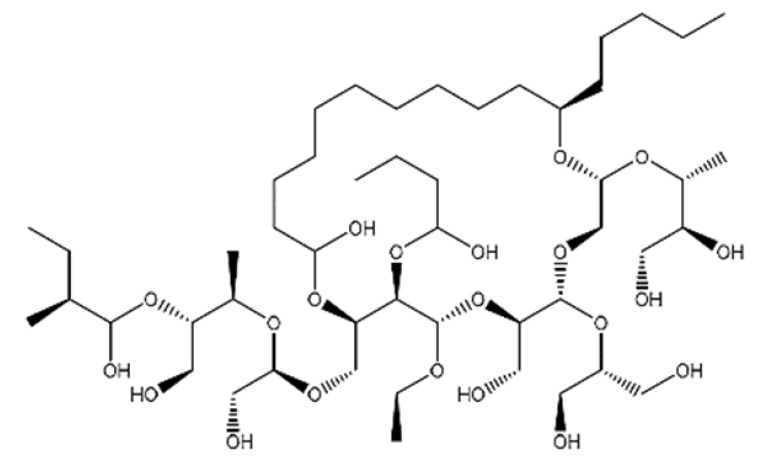	1008.55	n.p.	[66]

^1^ n.p.: not performed; ^2^ ER: efflux ratio.

## Data Availability

Not applicable.

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
