# Peer review of "Recent Advances on P-Glycoprotein (ABCB1) Transporter Modelling with In Silico Methods"

_ijms, 2022, doi:10.3390/ijms232314804_

Round 1
Reviewer 1 Report
In Lagares et al., the authors reviewed recent in-silico studies and progress in the P-glycoprotein (P-gp), which is an important modulator in multidrug resistance in the treatment of cancers and remain a popular drug target for several decades. This work first introduced the function and therapeutic significance of P-gp, and then provided a detailed overview of the major computational studies on P-gp over the last five years. The authors reviewed these studies which used both ligand- and structure-based methods, emphasized the impact of the rapid advances in machine learning techniques and structural biology on them, and discussed their perspectives on P-gp-related drug discovery.
Following a clear logic tread, the review manuscript provides a comprehensive overview of recent P-gp computational efforts. It will elucidate basic concepts and knowledge for newcomers and experienced scientists in this field. Overall, it meets the scope and standard of the International Journal of Molecular Sciences, especially in the special issue “ABC Transporters: Where Are We 45 Years On?”. Therefore, I recommend acceptance of this manuscript, after the minor issues described below are addressed:
1. The authors should add at least one figure as a “graphic abstract” or flowchart to summarize the major directions of in-silico P-gp studies (e.g. ligand-based vs structure-based, focused on protein-ligand binding or lipid function, etc.) Be sure to add an experimentally solved P-gp structure (X-ray of mouse P-gp or cryo-EM of human P-gp) to help newcomers to this field to understand its structure-function relationship better.
2. It would be extremely helpful if the authors could provide a table to summarize all the promising inhibitors for P-gp that were discovered by relevant computational studies reviewed in this manuscript, with their basic chemical properties (2D structure, molecular weight, polar surface area, etc.) as well as experimental validations (if exist) listed. This table would serve as an important guide and starting point for researchers who are interested in learning the most up-to-date P-gp inhibitor discovery progress and performing follow-up work such as lead optimization.
Author Response
- The authors should add at least one figure as a “graphic abstract” or flowchart to summarize the major directions of in-silico P-gp studies (e.g. ligand-based vs structure-based, focused on protein-ligand binding or lipid function, etc.) Be sure to add an experimentally solved P-gp structure (X-ray of mouse P-gp or cryo-EM of human P-gp) to help newcomers to this field to understand its structure-function relationship better.
A figure with the experimental human structures of the P-gp has been added at line 63. A new, more appropriate graphical abstract has also been provided.
- It would be extremely helpful if the authors could provide a table to summarize all the promising inhibitors for P-gp that were discovered by relevant computational studies reviewed in this manuscript, with their basic chemical properties (2D structure, molecular weight, polar surface area, etc.) as well as experimental validations (if exist) listed. This table would serve as an important guide and starting point for researchers who are interested in learning the most up-to-date P-gp inhibitor discovery progress and performing follow-up work such as lead optimization.
Thank you very much for this suggestion. Table 1 is added at line 495.
Reviewer 2 Report
Manuscript entitled “Recent advances on P-glycoprotein (ABCB1) transport model-2 ling with in silico methods”, is a great approach to compile the review data related to the particular domain with computational validation to hypothesize the outcome with support of computational results.
Although the review article is highly informative considering the different aspects related to the importance of drug resistance and involvement of ABC target I the development of antibiotic resistance, still I’m having some serious suggestions which can increase the quality and readability of the article.
1. Abstract should be more precise and informative to the concerned topic.
2. Full form of P-gp is not mentioned in the manuscript.
3. Introduction section of the manuscript I having too much information related to the computational drug design but not having much information related to the development of drug resistance and involvement of concerned macromolecular targets in the same.
4. Author’s have provided very less information about Three class classification models, which should be discussed in brief to increase its applicability.
5. What is the need to model the structure of human P-gp when it is already available in the pdb database?
6. What is the motive to execute the MD simulation analysis and for how much time period it has been executed?
7. What tools are employed for computational analysis were not mentioned in the manuscript.
8. Structure based drug design section is not clear as the authors are trying to review the work performed by other scientists or they are trying to execute the computational analysis by themselves?
9. Conclusion section is also not clear according to the discussed information in the manuscript.
10. Some of the recent citations related to the computational approaches for drug design needs to be updated in the manuscript:
· Jain R, Mujwar S. Repurposing metocurine as main protease inhibitor to develop novel antiviral therapy for COVID-19. Struct Chem. 2020 Dec 1;31(6):2487–99.
· Mujwar S, Harwansh RK. In silico bioprospecting of taraxerol as a main protease inhibitor of SARS-CoV-2 to develop therapy against COVID-19. Struct Chem. 2022 Oct 1;33(5):1517–28.
· Fidan O, Mujwar S, Kciuk M. Discovery of adapalene and dihydrotachysterol as antiviral agents for the Omicron variant of SARS-CoV-2 through computational drug repurposing. Mol Divers. 2022;
· Mujwar S, Sun L, Fidan O. In silico evaluation of food-derived carotenoids against SARS-CoV-2 drug targets: Crocin is a promising dietary supplement candidate for COVID-19. J Food Biochem. 2022 Sep 1;46(9).
· Shinu P, Sharma M, Gupta GL, Mujwar S, Kandeel M, Kumar M, et al. Computational Design, Synthesis, and Pharmacological Evaluation of Naproxen-Guaiacol Chimera for Gastro-Sparing Anti-Inflammatory Response by Selective COX2 Inhibition. Molecules 2022, Vol 27, Page 6905 [Internet]. 2022 Oct 14 [cited 2022 Oct 21];27(20):6905.
· Kciuk M, Mujwar S, Szymanowska A, Marciniak B, Bukowski K, Mojzych M, et al. Preparation of Novel Pyrazolo[4,3-e]tetrazolo[1,5-b][1,2,4]triazine Sulfonamides and Their Experimental and Computational Biological Studies. International Journal of Molecular Sciences 2022, Vol 23, Page 5892 [Internet]. 2022 May 24 [cited 2022 Oct 21];23(11):5892. Available from: https://www.mdpi.com/1422-0067/23/11/5892/htm
11. Authors can also explore riboswitch which has been already targeted in the past to counter the problems associated with the development of drug resistance.
· Mujwar, S. and Pardasani, K.R., Prediction of riboswitch as a potential drug target for infectious diseases: an insilico case study of anthrax, Journal of Medical Imaging and Health Informatics, Vol. 5(1), pp. 7-16, Feb 2015.
· Mujwar, S. and Pardasani, K.R., Prediction of riboswitch as a potential drug target and design of its optimal inhibitors for Mycobacterium tuberculosis. International Journal of Computational Biology and Drug Design Vol.8(4). 2015, 326-347.
Author Response
- Abstract should be more precise and informative to the concerned topic.
The abstract has been modified.
- Full form of P-gp is not mentioned in the manuscript.
A figure with the experimental human structures of the P-gp has been added in line 63.
- Introduction section of the manuscript I having too much information related to the computational drug design but not having much information related to the development of drug resistance and involvement of concerned macromolecular targets in the same.
Lines 52 to 57 added.
- Author’s have provided very less information about Three class classification models, which should be discussed in brief to increase its applicability.
It is not clear what should be added here. The reference No. 32 refers to the model, which is the only three class classification model that could be found in literature. It separates between inhibitors, substrates and non active compounds. It was developed in the author’s lab and is freely available on-line at https://www.vegahub.eu/portfolio-item/vega-qsar/. The domain of applicability is discussed in the reference 32 in datails, and also the on-line application within the VEGA Hub provides potential users with the information about their tested compounds, whether they are within or outside the model’s domain of applicability.
- What is the need to model the structure of human P-gp when it is already available in the pdb database?
I cannot say why other authors thought it was better to perform their study on homology model instead of the crystal structure but I assume some studies at the moment they were conducted the experimental structure was not yet available and it came out just at the moment when they already had the study completed the same way it happened to us. However a recent study1 has reported the use of homeology models in virtual screening applications with a superior performance in comparison to crystal structures. this fact can be explained by the confirmation of flexibility provided by homology models which allows a better accommodation of diverse ligands and therefore a better screening performance.
- What is the motive to execute the MD simulation analysis and for how much time period it has been executed?
Every presented study in this review uses a differet simulation time according to their goal and computational resources. As mentioned in the manuscript (Lines 362-453), the objectives are diverse from studying the involvement of membrane lipids in the transport mechanism to explore ligand binding interactions and the binding pocket of the protein.
- What tools are employed for computational analysis were not mentioned in the manuscript.
The computational analysis in the manuscript is referred to the specific case of P-gp and the computational studies that were conducted in the last five years. There might be other tools like pharmacophore modelling that were not included, because we narrowed down our mini-review to the last five years, when majority of studies have used machine learning approaches to build predictive models, and some also combine ligand-based machine learning approaches with structure-based techniques such as molecular docking or MD simulations. (Lines 134-137).
- Structure based drug design section is not clear as the authors are trying to review the work performed by other scientists or they are trying to execute the computational analysis by themselves?
It is a review paper. The studies that are presented are studies conducted by predominantely other scientists and we are reporting their most remarkable results. Some of our own studies in the recent years are also included, but not the on-going research work.
- Conclusion section is also not clear according to the discussed information in the manuscript.
Some lines have been added to the conclusion section (511 to 527)
- Some of the recent citations related to the computational approaches for drug design needs to be updated in the manuscript:
The references suggested are not related to ABC transporters or to P-glycoprotein which is the main topic of the review paper. The references regarding computational approaches included in the paper are references that talk just about computational tools specifically.
- Jain R, Mujwar S. Repurposing metocurine as main protease inhibitor to develop novel antiviral therapy for COVID-19. Struct Chem. 2020 Dec 1;31(6):2487–99.
- Mujwar S, Harwansh RK. In silico bioprospecting of taraxerol as a main protease inhibitor of SARS-CoV-2 to develop therapy against COVID-19. Struct Chem. 2022 Oct 1;33(5):1517–28.
- Fidan O, Mujwar S, Kciuk M. Discovery of adapalene and dihydrotachysterol as antiviral agents for the Omicron variant of SARS-CoV-2 through computational drug repurposing. Mol Divers. 2022;
- Mujwar S, Sun L, Fidan O. In silico evaluation of food-derived carotenoids against SARS-CoV-2 drug targets: Crocin is a promising dietary supplement candidate for COVID-19. J Food Biochem. 2022 Sep 1;46(9).
- Shinu P, Sharma M, Gupta GL, Mujwar S, Kandeel M, Kumar M, et al. Computational Design, Synthesis, and Pharmacological Evaluation of Naproxen-Guaiacol Chimera for Gastro-Sparing Anti-Inflammatory Response by Selective COX2 Inhibition. Molecules 2022, Vol 27, Page 6905 [Internet]. 2022 Oct 14 [cited 2022 Oct 21];27(20):6905.
- Kciuk M, Mujwar S, Szymanowska A, Marciniak B, Bukowski K, Mojzych M, et al. Preparation of Novel Pyrazolo[4,3-e]tetrazolo[1,5-b][1,2,4]triazine Sulfonamides and Their Experimental and Computational Biological Studies. International Journal of Molecular Sciences 2022, Vol 23, Page 5892 [Internet]. 2022 May 24 [cited 2022 Oct 21];23(11):5892. Available from: https://www.mdpi.com/1422-0067/23/11/5892/htm
- Authors can also explore riboswitch which has been already targeted in the past to counter the problems associated with the development of drug resistance.
Again I don't consider these manuscripts relevant for the review paper because the main topic of the review is not drug resistance, it is computational tools used to study the P-gp.
- Mujwar, S. and Pardasani, K.R., Prediction of riboswitch as a potential drug target for infectious diseases: an insilico case study of anthrax, Journal of Medical Imaging and Health Informatics, Vol. 5(1), pp. 7-16, Feb 2015.
- Mujwar, S. and Pardasani, K.R., Prediction of riboswitch as a potential drug target and design ofits optimal inhibitors for Mycobacterium tuberculosis. International Journal of Computational Biology and Drug Design Vol.8(4). 2015, 326-347.
- References:
- Mordalski, S.; Witek, J.; Smusz, S.; Rataj, K.; Bojarski, A.J. Multiple conformational states in retrospective
virtual screening–homology models vs. crystal structures: Beta-2 adrenergic receptor case study. J. Cheminform.
2015, 7, 13.
Round 2
Reviewer 2 Report
Authors have resolved most of the comments and now the manuscript seems to be in acceptable form.